# Predictors of Viral Non-Suppression among Patients Living with HIV under Dolutegravir in Bunia, Democratic Republic of Congo: A Prospective Cohort Study

**DOI:** 10.3390/ijerph19031085

**Published:** 2022-01-19

**Authors:** Roger T. Buju, Pierre Z. Akilimali, Erick N. Kamangu, Gauthier K. Mesia, Jean Marie N. Kayembe, Hippolyte N. Situakibanza

**Affiliations:** 1Department of Public Health, Faculté de Medicine, University of Bunia, Bunia P.O. Box 292, Congo; rogerbujutsedha1@gmail.com; 2Department of Biostatistics and Epidemiology, Kinshasa School of Public Health, University of Kinshasa, Kinshasa P.O. Box 11850, Congo; 3Département des Sciences de Base, School of Medicine, University of Kinshasa, Kinshasa P.O. Box 11850, Congo; erick.kamangu@unikin.ac.cd; 4Département de Pharmacologie Clinique, School of Medicine, University of Kinshasa, Kinshasa P.O. Box 11850, Congo; mesia.kahunu@unikin.ac.cd; 5Department Internal Medicine, School of Medicine, University of Kinshasa, Kinshasa P.O. Box 11850, Congo; jm.kayembe@unikin.ac.cd; 6Department of Tropical Medicine and Infectious Disease, School of Medicine, University of Kinshasa, Kinshasa P.O. Box 11850, Congo; situakibanza.nani@unikin.ac.cd

**Keywords:** dolutegravir, virologic failure, conflict setting, Bunia

## Abstract

The Democratic Republic of the Congo adopted the integrase inhibitor dolutegravir (DTG) as part of its preferred first-line HIV treatment regimen in 2019. This study aimed to identify predictors of viral non-suppression among HIV-infected patients under a DTG-based regimen in the context of ongoing armed conflict since 2017 in the city of Bunia in the DRC. We conducted a cohort study of 468 patients living with HIV under DTG in all health facilities in Bunia. We calculated the proportion of participants with an HIV RNA of below 50 copies per milliliter. About three in four patients (72.8%) in this cohort had a viral load (VL) of <50 copies/mL after 6–12 months. After controlling for the effect of other covariates, the likelihood of having non-suppression remained significantly lower among the 25–34 age group and self-reported naïve patients with a baseline VL of ≥50 copies/mL. The likelihood of having non-suppression remained significantly higher among those who were at advanced stages of the disease, those with abnormal serum creatinine, those with high baseline HIV viremia over 1000 copies/mL, and the Sudanese ethnic group compared to the reference groups. This study suggests that we should better evaluate adherence, especially among adolescents and economically vulnerable populations, such as the Sudanese ethnic group in the city of Bunia. This suggests that an awareness of the potential effects of DTG and tenofovir is important for providers who take care of HIV-positive patients using antiretroviral therapy (ART), especially those with abnormal serum creatinine levels before starting treatment.

## 1. Introduction

Even with the discovery of treatment, HIV infection remains a major public health problem in the world. Despite the proven efficacy of triple therapy with tenofovir, lamivudine, and dolutegravir (TDF + 3TC + DTG), predicting resistance and adverse events in patients on this regimen is a necessity to monitor virological failure (VF) [1,2], especially in resource-limited countries stricken by political and security instability. Antiretroviral therapy (ART) is aimed at achieving an undetectable viral load (VL) (<50 copies/mL) within 6 months of treatment initiation. VF is defined as a VL of ≥50 copies/mL 6 months after ART initiation [3]. Antiretroviral (ART) resistance is related to the selection of mutants responsible for viral replication in the presence of ART [4,5]. This selection depends on pharmacological factors, the efficacy of the antiviral treatment, and the genetic barrier of the virus to different ARTs [6]. Previous studies reported that poor adherence to ART, WHO clinical stages III and IV, level of education, co-infection, and certain substances such as nicotine in tobacco, alcohol, and drugs were the main predictors of ART resistance [7,8,9,10,11,12]. A group of researchers has shown that non-nucleoside reverse transcriptase inhibitors (NNRTI) resistance ultimately appears to predict treatment failure in people living with HIV/AIDS (PLHIV) at the start of ART with DTG [13].

Bunia is a conflict setting, and dolutegravir (DTG) was introduced there in July 2019. The standard first line of ART initiation used before included the combination of Tenofovir (TDF), Lamivudine (3TC), and Efavirenz (EFV) (TDF + 3TC + EFV) as first-line treatment, followed by Zidovudine (AZT), Lamivudine (3TC), and Nevirapine (NVP) (AZT + 3TC + NVP), and the combination of Tenofovir (TDF) with Lamivudine (3TC) (TDF + 3TC) in combination with Nevirapine (NVP) as an alternative regimen. Conflict-affected and displaced populations face major challenges in terms of access to health facilities, food security, and good adherence to ART and treatment outcomes [14,15]. For HIV services located in a (post-)conflict setting, long distances to health facilities are not only a barrier due to the costs incurred by accessing the service but also due to the great risk taken by traveling in unsafe areas. Bunia city, a city experiencing political and security instability, has weak technical facilities, and the displacement of the population, including HIV patients, means that we need to assess the advantages of introducing DTG into their regimen. Despite the factors studied in the context of stability, in the literature review, we did not find any studies analyzing the process of VL suppression in the context of armed conflict, especially under DTG.

Studies have shown that DTG develops a resistance of about 1% and can prevent the kidneys from releasing creatinine waste products in the urine without impacting renal function [16,17,18]. Thus, a small increase of about 10 μmol/L and a persistent creatinine level is observed under this regimen. In addition, a significantly elevated concentration of DTG without explanation was observed in HIV patients aged 60 and older [19]. This study aimed to identify predictors of viral failure (or viral non-suppression) among HIV-infected patients under a DTG-based regimen in the context of ongoing armed conflict since 2017 in the city of Bunia in the Ituri province, Democratic Republic of the Congo (DRC).

## 2. Methods

### 2.1. Study Design and Participants

Between July 2019 and July 2021, we conducted an observational prospective cohort study of 468 patients living with HIV under DTG in all health facilities in Bunia. Bunia is a city located in the eastern part of the DRC. The city has seen armed conflict since 2017, and there are still areas where the conflict continues to this day. We included in this study patients who were aged 18 or older.

### 2.2. Procedures, Data Collection, and Outcome

Upon inclusion, participants were switched to a regimen of DTG (50 mg), lamivudine (300 mg), and tenofovir disoproxil fumarate (300 mg), which were all taken orally once daily. Thereafter, scheduled visits to assess the patients’ viral loads were performed at six-month intervals. During all visits, participants underwent medical evaluation, including physical examination, reporting of adverse events, review of concomitant medications, as well as HIV-1 RNA viral load, hemogram, liver, urine, and renal function tests.

The questionnaire was designed to gather sociological and ethnical data from the participants. Trained interviewers from the research team conducted the data collection. After pre-testing the first 10 patients, the questions were checked for consistency and modified as necessary. Designated supervisors monitored the data collection to ensure the validity and consistency of the data, as well as compliance with ethical guidelines.

The following information was collected: socio-demographic characteristics (age, gender, marital status, education, and residence), clinical characteristics (status of treatment before enrollment, tobacco consumption, WHO HIV clinical stage, exposition time under the DTG-based regimen in months), and biological characteristics (serum creatinine at baseline). Exposition time under the DTG-based regimen in months is the duration from initiation of the DTG-based regimen to the day when the blood samples were collected to assess viral loads. Serum creatinine values above 1.5 mg/dL in men and above 1.37 mg/dL in women were considered high (abnormal).

The primary outcome was the proportion of patients with a viral load below 50 copies per milliliter after 6 months. Not all patients were able to take a viral load sample after six months. Virological suppression was defined as an HIV RNA viral load of less than 50 copies/mL [3]. Status of treatment before enrollment was defined in three categories: experienced patients, self-reported naïve patients with a baseline VL of <50 copies/mL, and self-reported naïve patients with a baseline VL of ≥50 copies/mL.

### 2.3. Statistical Analysis

We analyzed the proportion of participants with an HIV RNA below 50 copies per mL. Since viral load is influenced by the duration of treatment, we adjusted the viral response according to the duration of treatment under the DTG-based regimen.

Descriptive analysis of the individuals’ characteristics was performed using frequency tables for the categorical variables and for the mean and standard deviation of the patients’ ages. Differences in socio-demographic and clinical characteristics according to prior history of ART (ART naïve vs. those who were previously under ART) were assessed using the t-test for continuous variables and the chi-squared test for the independence of the categorical variables.

We compared viral suppression by referring to the background characteristics of the patients under DTG using the chi-squared test. We used a logistic regression model to identify predictors of virological non-suppression. Gender, age, education, marital status, ethnic group, residence, serum creatinine levels, stage of disease, tobacco history, viral load at baseline, status of treatment before enrollment, and time under the DTG-based regimen in months were introduced into a logistic model to identify the predictors of viral non-suppression. Multicollinearity was assessed using variance inflation factors (VIF) greater than 4.0 and found to be insignificant—the mean VIF was 1.91. All tests were two-sided, and the level of significance was set to *p* < 0.05. All tests were performed using Stata software (version 14.0, Stata Corporation, College Station, TX, USA).

### 2.4. Ethical Statement

The study protocol was approved by the Institutional Review Board of the Ethics Committee for Research Subjects at the Kinshasa University School of Public Health (No App: ESP/CE/094/2018 of 9 August 2018). All participants provided written informed consent prior to participation in the study. However, patients’ records/information were anonymized and de-identified prior to the analysis.

## 3. Results

The background characteristics of the participants are shown in Table 1. This study included 468 patients living with HIV with a mean age of 38.97 (SD = 11.94); of these, 7 in 10 were female. In general, the level of education was low, with one in three having attended or completed high school. About one-fourth (43.8%) reported that they were married or living with a partner. Five percent were living in a rural setting, while eight percent were ethnically Sudanese. One-third (33.8%) had an abnormal serum creatinine level. Half of the participants were at an advanced stage of the disease (Stage III and IV) (53.2%).

Figure 1 shows that 62.2% of the patients were under treatment before their inclusion in the study (experienced patients). One-fifth were self-reported naïve patients with a baseline VL of <50 copies/mL, while 18% were self-reported naïve patients with a baseline VL of ≥50 copies/mL.

In this cohort, more than one-fourth (28.8% (135)) were lost to follow-ups (LTFU) during the study period, and the mortality rate was 5.0% (Figure 2). A total of 3435.22 person-months (p-m) were involved in follow-ups, with an overall incidence rate of 5.82 deaths per 1000 p-m and 33.48 LTFU per 1000 p-m. As shown in Figure 3, 12%(56), 21.4% (100), 26.5% (124), 28.6% (134), and 28.8% (135) of the patients were lost to follow-ups after 1, 3, 6, 9, and 12 months, respectively.

The baseline characteristics of the individuals with available viral loads are presented in Table 2.

There were 305 records that included viral load results at 6–12 months. Nearly three in four patients were virally suppressed at 6 to 12 months (72.8%). About one-fourth (27.2%; 95% CI: from 22.5 to 32.5) of the participants had a VL of ≥50 copies/mL after the follow-up period. The proportion of participants with a VL of ≥50 copies/mL was highest among the Sudanese ethnic group, with about 55.6% of these individuals failing viral suppression. Patients with abnormal serum creatinine levels had a higher likelihood of having a VL of ≥50 copies/mL compared to those who had normal serum creatinine levels. The viral response was significantly lower among participants at an advanced stage of the disease (Stage III and IV; 32.9% of non-suppression cases). Although they had a viral load of less than 50 copies/mL at baseline, 30.8% of the self-reported naïve patients with a baseline VL of <50 copies/mL had a viral load over 50 copies/mL at 6–12 months after the initiation of the DTG-based regimen.

Figure 4 shows that, among the experienced patients who had a VL < 1000 copies/mL, about one in ten (9.4%) experienced a rebound of viral load.

Table 3 shows the predictors of non-viral suppression in a multivariate logistic regression model. After controlling for the effect of other covariates, the likelihood of non-suppression remained significantly lower among adults, especially among those in the 25–34 age group (AOR  =  0.33, 95% CI 0.12–0.93) and the self-reported naïve patients with a baseline VL of ≥50 copies/mL compared to their reference group. The likelihood of non-suppression remained significantly higher among those with abnormal serum creatinine levels (AOR  =  2.32, 95% CI 1.22–4.43), those who were at an advanced stage of the disease (stage III and IV) (AOR  =  1.86, 95% CI 1.01–3.43), those who had a high baseline HIV viremia of over 1000 copies/mL (AOR  =  3.41, 95% CI 1.64–7.08), and the ethnically Sudanese population (AOR  =  4.19, 95% CI 1.43–12.68) compared to their reference groups.

## 4. Discussion

This study is the first in the DRC to examine predictors of non-suppression among PLHIV on DTG in Bunia, a city that has been experiencing the ongoing effects of armed conflict for more than five years. About three in four patients (72.8%) in this cohort had a VL of <50 copies/mL after 6–12 months. A previous study reporting viral suppression used a cut-off VL of ≥1000 copies/mL (76.8% of the participants had achieved viral suppression) [20]; however, the non-suppression rate reported in this study (74%) is similar to the rate reported by Calmy et al. [21]. Regarding predictors of non-suppression after initiating the DTG-based regimen, the current study revealed that patients with abnormal serum creatinine levels, those who were at an advanced stage of the disease (stage III and IV), those with a high baseline HIV viremia over 1000 copies/mL, and the Sudanese ethnic group were found to have increased odds of non-suppression (VL ≥ 50 copies/mL) in this cohort study. However, the likelihood of having non-suppression remained significantly lower among the 25–34 age group and self-reported naïve patients with baseline VL ≥ 50 copies/mL.

In this study, patients with abnormal serum creatinine levels at baseline were more likely to have a viral load of ≥50 copies/mL compared to those with normal serum creatinine levels at enrollment. In a previous study, DTG caused a significant decrease in the glomerular filtration rate in PLHIV without changes in the other renal markers studied, despite the presence of tenofovir [22]. Another study showed that, after starting the DTG regimen, few patients showed increases in serum creatinine from a baseline of >50% or 0.5 mg/dL, and no discontinuations due to chronic kidney disease (CKD) were documented [23]. Another group of investigators observed a low prevalence of chronic kidney disease in Asian people; however, progression to CKD was substantially associated with tenofovir exposure [24]. In the present study, patients were also administered tenofovir. Tenofovir disoproxil fumarate has been associated with elevated serum creatinine levels and renal toxicity [25], which necessitates renal function monitoring, especially for those with abnormal serum creatinine levels before starting treatment. Patients with high serum creatinine levels received the same dose of DTG as the other patients in this cohort. Renal failure would increase the half-life of ART and make patients with abnormal serum creatinine levels more prone to side effects. Side effects are one of the reasons for reduced adherence, inadequate follow-ups, low self-efficacy, low treatment satisfaction to ART, and consequently, poor viral response compared to patients with normal renal function [26,27,28,29]. The authors could not certify if the individuals enrolled with abnormal serum creatinine levels were in the acute phase or not.

As expected, the viral response was poorest in patients with advanced disease (stages III and IV). Our results are consistent with a previous study showing that patients with advanced disease are more likely to have side effects, including a detectable viral load [30]. Unfortunately, we did not monitor/report side effects associated with DTG use. In addition, we found that a high baseline HIV viral load was associated with an increased risk of non-suppression.

The probability of having viral non-suppression remained significantly lower in older patients compared to adolescents, particularly among those in the 25–34 age group. As previous studies have also reported, the likelihood of having viral non-suppression is significantly lower in adolescents [31,32,33]. Poorer adherence among adolescents and increasing treatment experience among older long-term survivors could explain this high rate of non-suppression among adolescents. This low proportion of viral suppression among adolescents in Bunia can also be explained by the security instability brought about by armed conflict, which disrupts ART retention programs for people living with HIV/AIDS. Young adolescents are the most involved and recruited in armed conflicts, and young girls are used as sexual objects. A previous study conducted in the context of armed conflict has shown that, overall, in the event of armed conflict, the destruction of health infrastructure can destabilize ART programs and lead to interruptions in treatment and the supply chain of inputs and ART, the displacement of staff and civilians, and the demoralization of the population leading to non-retention in the ART program [34]. However, most studies on viral load suppression worldwide have not reached these results under DTG, which has been shown to be highly effective at rapidly suppressing viral load [35]. A study in sub-Saharan Africa showed that armed conflict leads to widespread migration and refugees, the proliferation of AIDS, weakening property rights, and restricted access to basic social services [36]. Overall, we believe that the loss of retention due to armed conflict and social unrest is responsible for the failure to eliminate viral load, even under DTG, which suppresses viral loads below limit of detection in most individuals within 6 months [37].

The other group of patients with a poor viral response were the Sudanese ethnic group. The proportion of participants with a viral load of ≥50 copies/mL was about 55% among the Sudanese ethnic group, and the probability of viral non-suppression remained significantly higher among them than among other ethnicities. We believe that the socio-economic and cultural situation of the Sudanese population living in the city of Bunia, coupled with the disruption of economic accessibility for the majority of them [38,39], exposes them to food insecurity; food insecurity is associated with poor adherence to treatment.

This study has several limitations. First is the possibility of misclassifying the patients’ status of treatment before enrollment, which the participants may have incorrectly disclosed. Some patients were self-declared naïve when they had an undetectable viral load. Some patients on ART who changed treatment sites did not disclose their treatment status (former ART case) to the new providers for fear of being stigmatized. In a previous study, 19.4% of patients on ART did not disclose their status to family and friends [40]. The study of individuals not disclosing such information was performed in Nepal, which is probably a different setting from the DRC.

It is difficult to testify about the viral rebound among patients who were self-declared naïve but had a viral load of less than 50 copies/mL and who ended up with a viral load of more than 50 copies after 6 months under the new regimen. Patients who did not disclose their HIV status with their family in the previous study had a higher likelihood of being lost to follow-ups [41]. Thus, patients who do not disclose their status are often less likely to receive social support and will perform poorly in terms of achieving optimum levels of adherence and retention in care. In the context of a post-conflict setting, where basic social infrastructure might be in disarray and poverty often prevails, social support from the family and community is vital to foster the engagement of a patient in HIV treatment and care [41]. Secondly, due to resource constraints, not all patients had the 6 months’ VL when expected; some did have viral loads at 7, 8, 9, 10, and even 12 months. We would have liked the collection of the blood samples, used to assess viral load after the initiation of the DTG-based regimen, to have been performed after a fixed duration of exposure (six months) for all participants. We decided to include them all and adjust for the duration under DTG to increase the power of our analysis. This study included a low number of individuals from the Sudanese ethnic group. These results should be evaluated with caution. More individuals of this ethnic group should be analyzed to support our results. Another potential limitation of this study was that the research team could not ascertain the viral load response of the participants who were LTFU. Finally, although we have used multivariate regression, we cannot exclude the potential confounding effect of the other variables, such as daily adherence, on the patients’ viral load response.

## 5. Conclusions

This study suggests that we should better evaluate adherence to ART treatment in HIV patients, especially among adolescents and economically vulnerable populations, such as the Sudanese ethnic group in the city of Bunia. This also suggests that an awareness of the potential effect of DTG and tenofovir is important for providers who take care of HIV-positive patients who are on ART, especially those with abnormal serum creatinine levels before starting treatment.

## Figures and Tables

**Figure 1 ijerph-19-01085-f001:**
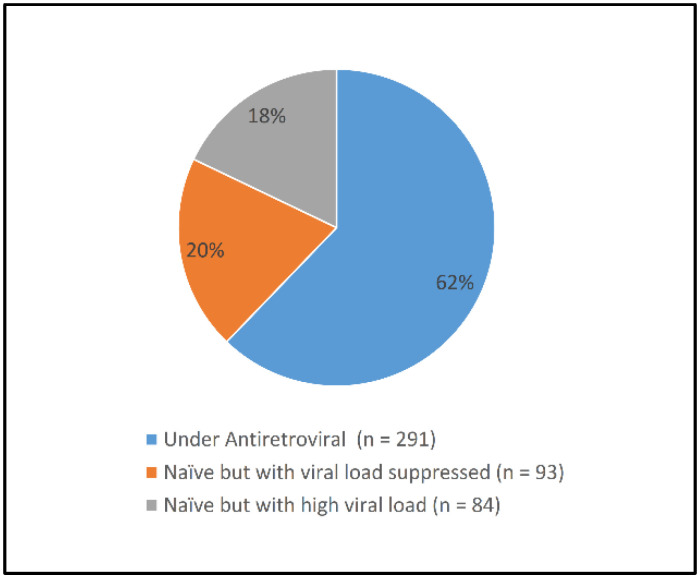
Status of patients at enrollment.

**Figure 2 ijerph-19-01085-f002:**
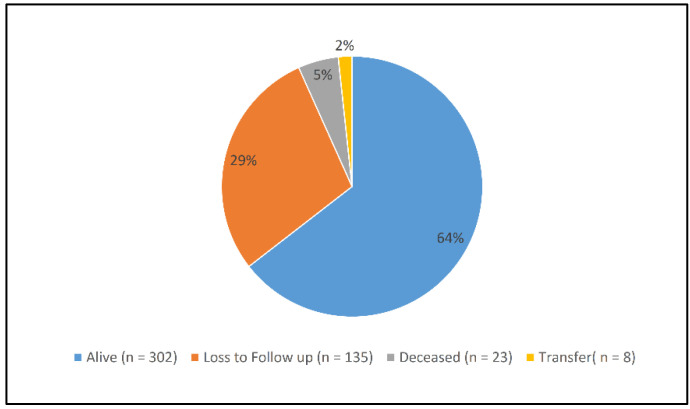
Status of patients at the end of the study.

**Figure 3 ijerph-19-01085-f003:**
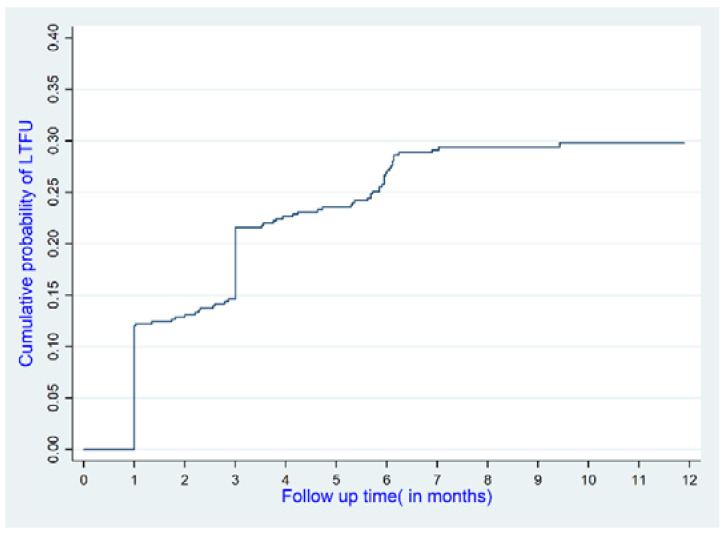
Incidence of Loss to follow up.

**Figure 4 ijerph-19-01085-f004:**
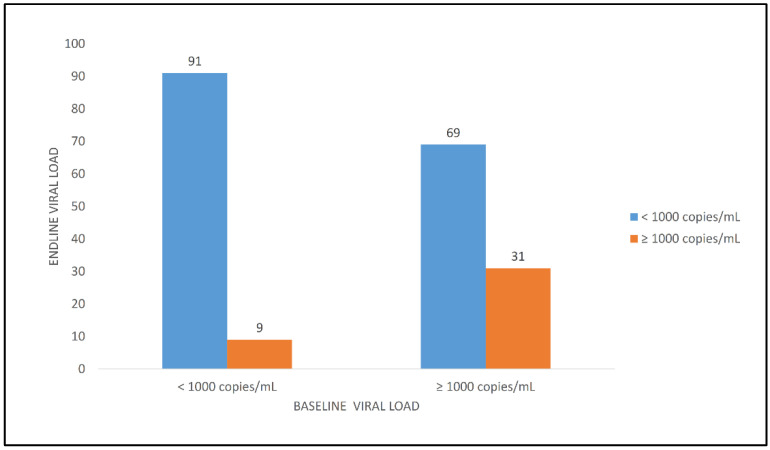
Change of viral load among experienced patients (*n* = 220).

**Table 1 ijerph-19-01085-t001:** Background characteristics.

	Naïve Patients		Under ART (Experienced)		Overall		*p*-Value
	*n*	%	*n*	%	*n*	%	
Age	36.74 ± 11.72		40.32 ± 11.89		38.97 ± 11.94		<0.001
Age							0.007
18–24	21	11.9	24	8.2	45	9.6	
25–34	65	36.7	73	25.1	138	29.5	
35–49	67	37.9	129	44.3	196	41.9	
50–73	24	13.6	65	22.3	89	19.0	
Sex							0.065
Female	114	64.4	211	72.5	325	69.4	
Male	63	35.6	80	27.5	143	30.6	
Education							0.983
None/Primary	116	65.5	191	65.6	307	65.6	
Secondary/Tertiary	61	34.5	100	34.4	161	34.4	
Marital status							0.151
Living alone	92	52.0	171	58.8	263	56.2	
In Union	85	48.0	120	41.2	205	43.8	
Ethnic group							0.653
Nilotic	79	44.6	145	49.8	224	47.9	
Bantu	48	27.1	66	22.7	114	24.4	
Semi-Bantu	35	19.8	58	19.9	93	19.9	
Sudanese	15	8.5	22	7.6	37	7.9	
Residence							0.052
Urban	173	97.7	273	93.8	446	95.3	
Rural	4	2.3	18	6.2	22	4.7	
Creatinine							0.039
Normal	107	60.5	203	69.8	310	66.2	
Abnormal	70	39.5	88	30.2	158	33.8	
Stage of disease							0.068
Stage I	43	24.3	92	31.6	135	28.8	
Stage II	26	14.7	58	19.9	84	17.9	
Stage III	87	49.2	116	39.9	203	43.4	
Stage IV	21	11.9	25	8.6	46	9.8	
Tobacco consumption							0.177
No	129	72.9	228	78.4	357	76.3	
Yes	48	27.1	63	21.6	111	23.7	
Total *	177	37.8	291	62.2	468	100.0	

*: percentage calculated among 468 participants. ART: antiretroviral therapy.

**Table 2 ijerph-19-01085-t002:** Background characteristics of patients living with HIV under dolutegravir with and without virological suppression.

	N	More than 50 Copies/mL		Less than 50 Copies/mL		*p*-Value
		*n*	%	*n*	%	
Overall	305	83	27.2	222	72.8	
Age						0.140
18–24	29	10	34.5	19	65.5	
25–34	87	16	18.4	71	81.6	
35–49	133	42	31.6	91	68.4	
50–73	56	15	26.8	41	73.2	
Sex						0.688
Female	219	61	27.9	158	72.1	
Male	86	22	25.6	64	74.4	
Education						0.802
None/Primary	195	54	27.7	141	72.3	
Secondary/Tertiary	110	29	26.4	81	73.6	
Marital status						0.435
Living alone	180	46	25.6	134	74.4	
In Union	125	37	29.6	88	70.4	
Ethnic group						0.0106
Nilotic	157	34	21.7	123	78.3	
Bantu	72	22	30.6	50	69.4	
Semi-Bantu	58	17	29.3	41	70.7	
Sudanese	18	10	55.6	8	44.4	
Residence						0.362
Urban	288	80	27.8	208	72.2	
Rural	17	3	17.6	14	82.4	
Creatinine						0.002
Normal	233	53	22.7	180	77.3	
Abnormal	72	30	41.7	42	58.3	
Stage						0.033
I and II	159	35	22.0	124	78.0	
III and IV	146	48	32.9	98	67.1	
Tobacco consumption						0.540
No	239	67	28.0	172	72.0	
Yes	66	16	24.2	50	75.8	
Status of treatment at baseline						0.814
Under ART before inclusion	220	58	26.4	162	73.6	
Self-reported naïve patients with baseline VL < 50 copies/mL	52	16	30.8	36	69.2	
Self-reported naïve patients with baseline VL ≥ 50 copies/mL	33	9	27.3	24	72.7	
Viral load at baseline						0.002
<1000 copies/mL	224	50	22.3	174	77.7	
≥1000 copies/mL	81	33	40.7	48	59.3	
Exposition time under DTG-based regimen in months						0.057
6 to 7	98	26	26.5	72	73.5	
8 to 9	49	7	14.3	42	85.7	
10 to 12	158	50	31.6	108	68.4	

**Table 3 ijerph-19-01085-t003:** Predictors of virological non-suppression among patients living with HIV under dolutegravir (*n* = 305) *.

	N	Crude			Adjusted		
		OR	95% CI	*p*-Value	OR	95% CI	*p*-Value
Age							
18–24	29	1			1		
25–34	87	0.43	0.17–1.09	0.076	0.33	0.12–0.93	0.035
35–49	133	0.88	0.37–2.05	0.762	0.69	0.25–1.85	0.455
50 and +	56	0.70	0.26–1.83	0.461	0.53	0.17–1.61	0.259
Sex							
Female	219	1			1		
Male	86	0.89	0.50–1.57	0.688	0.94	0.47–1.87	0.854
Education							
None/Primary	195	1			1		
Secondary/Tertiary	110	0.93	0.55–1.58	0.802	1.29	0.67–2.48	0.446
Marital status							
Living alone	180	1			1		
In Union	125	1.22	0.74–2.04	0.435	1.28	0.68–2.42	0.447
Ethnic group							
Nilotic	157	1			1		
Bantu	72	1.59	0.85–2.99	0.147	180	0.79–3.34	0.122
Semi-Bantu	58	1.50	0.76–2.96	0.243	1.51	0.67–2.93	0.284
Sudanese	18	4.52	1.66–12.34	0.003	4.19	1.43–12.68	0.012
Residence							
Rural	17	1			1		
Urban	288	0.56	0.16–1.99	0.368	0.62	0.14–2.76	0.527
Creatinine							
Normal	233	1			1		
Abnormal	72	2.43	1.39–4.25	0.002	2.32	1.22–4.43	0.010
Stage							
I and II	159	1			1		
III and IV	146	1.74	1.04–2.89	0.034	1.86	1.01–3.43	0.047
Viral load at baseline							
<1000 copies/mL	224	1			1		
≥1000 copies/mL	81	2.39	1.39–4.12	0.002	3.41	1.64–7.08	0.001
Tobacco consumption							
No	239	1			1		
Yes	66	0.82	0.43–1.54	0.541	0.79	0.35–1.78	0.565
Status of treatment at baseline							
Under ART before inclusion	220	1			1		
Self-reported naïve patients with baseline VL < 50 copies/mL	52	1.24	0.521	0.64–2.40	1.44	0.66–3.12	0.361
Self-reported naïve patients with baseline VL ≥ 50 copies/mL	33	1.05	0.912	0.46–2.38	0.32	0.11–0.92	0.034
Exposition time under DTG-based regimen in months							
6 to 7	98	1			1		
8 to 9	49	0.46	0.18–1.15	0.099	0.71	0.26–1.96	0.510
10 to 12	158	1.28	0.73–2.24	0.384	1.22	0.64–2.33	0.554

*: Those who had viral load results at 6–12 months.

## Data Availability

Data are available upon request to pierretulanefp@gmail.com.

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
