# Peer review of "Predictors of Viral Non-Suppression among Patients Living with HIV under Dolutegravir in Bunia, Democratic Republic of Congo: A Prospective Cohort Study"

_ijerph, 2022, doi:10.3390/ijerph19031085_

Round 1

Reviewer 1 Report

This manuscript reports viral suppression in a mostly urban HIV-positive population living in a conflict zone in the Democratic Republic of Congo who was treated with dolutegravir, as per new guidelines. The authors investigated factors, including ethnicity, associated with viral non-suppression (or failure – see below). This paper is timely and important as dolutegravir roll-out is accelerating in African countries.

Major criticisms:

Some outcomes should be reported as time dependent.

As it is currently presented, the data does not permit to distinguish between non-suppression and treatment failure. The authors should report both outcomes separately. If needed, and depending on data available to the authors, failure might be defined as a detectable viral load after viral load suppression (if appropriate this will need to be included in the methods). The authors should include a survival curve reporting the odds of achieving viral suppression over time for patients with VL available or use cumulative incidence rate of achieving viral suppression.

The same applies to results presented in Figure 2 where follow-up outcomes are presented without indication of the duration of follow-up. It is essential to present this information in a time-dependent manner (i.e., what percentage of patients discontinued over time; what proportion of patients were lost to follow-up after 6, 12 months; etc.…). Survival curves may be used.

Overall viral suppression rate should be reported in the abstract as it is the primary outcome of this study.

Lower likelihood of viral suppression was not observed in older adults but only in the 25-34 group. Thus, the conclusion page 7, in the abstract and in the first paragraph of the discussion needs to be corrected.

The 1,000 copies cut-off used by the authors for high vs. low viral loads is not the most appropriate. They need to add more viral load categories in a sub-analysis to clarify their observation.

As explained above, the following statement page 9: “… significantly lower in older patients…” is untrue. From data presented in Table 2, only the 25-34 group had significantly lower chances of non-suppression compared to 15-24 group. In addition, the youngest age group should be changed to 18-24 since the authors stated in the methods that inclusion was limited to adult above 18 years old. If minors were included, changes to the methods (including in ethical statements) should be made.

Minor comments:

Page 2, in the description of the regimen, tenofovir should be replaced by tenofovir disoproxil fumarate.

Page 2: “Some participants achieved…” This statement should be in the Results section rather than the Methods. Same for page 3 for the same statement.

Page 2: “We used a semi-structured questionnaire…” The authors should specify that (I assume) the questionnaire was designed to gather sociological and ethnical data from the participants.

Same section: “After pre-testing, the tools were…” This section is very vague. How many participants were enrolled in the pre-testing phase? What outcome was evaluated in this pre-testing phase (i.e., participation, satisfaction?). Do “tools” in fact mean questions included in the questionnaire?

Page 3: “The primary objective…” The current study is an observational one. As such, the objective cannot be to achieve viral suppression (that would imply an interventional study). Rather the authors should indicate that their primary outcome was the proportion of patients with viral loads below 50 RNA copies per ml of plasma. Secondary outcome was identifying socio-economical and clinical factors associated with viral failure (as defined as viral load >50 RNA copies / ml at follow-up).

Can the authors comment on patients who self-reported as treatment naïve with viral suppression?

Figure 1 is useless as this information is already included in Table 1.

Page 5: “changes in viral load” are actually not presented in Table 2. It would be more accurate to state that baseline characteristics of individuals with available viral loads are presented in Table 2.

Table 2. It may be better to present overall results at the top of the table.

“Residence” should be moved to the left of the table.

The definition for “normal” vs. “abnormal” serum creatinine levels should be clearly indicated in the methods.

The authors need to discuss what in their opinion explains why individuals with higher creatinine levels were at higher chance of non-suppression. Is that because of discontinuation from medical indication? I expect not but the authors need to discuss this specific point rather than listing previous observations (that themselves do not provide an explanation for non-suppression).

Page 9, second paragraph. The authors did not monitor/report side effects associated with dolutegravir use. Thus, the 2nd sentence should be edited as it refers to side effects.

Page 9 “which removes viral load” is inaccurate. It would be better to state that dolutegravir-based therapy typically suppresses viral loads below limit of detection in most individuals within 6 months (and add references).

Page 9, the sentence starting with “Whereas it is…” is incomplete. Please edit.

Author Response

  1. Some outcomes should be reported as time dependent.

We have now reported mortality and Loss to follow up as time dependent. Even if we plan a full paper analysing the two outcomes.

  1. As it is currently presented, the data does not permit to distinguish between non-suppression and treatment failure. The authors should report both outcomes separately. If needed, and depending on data available to the authors, failure might be defined as a detectable viral load after viral load suppression (if appropriate this will need to be included in the methods). The authors should include a survival curve reporting the odds of achieving viral suppression over time for patients with VL available or use cumulative incidence rate of achieving viral suppression.

This paper focussed on non-suppression VL. Some of Patient were naive patients and treatment failure was not our main objective. We have reported cumulative incidence rate of achieving viral suppression which was stated in the manuscript as follow:  Nearly three in four patients were virally suppressed at 6 to 12 months (73%). However, we have added the proportion of VL rebound among experienced patients who had baseline VL<1000 copies/ml, this can be read as follow: «Figure 4 is showing that among experienced patients who had a VL<1000 copies/ml, about one in ten (9.4%) experienced a rebound of viral load. »

  1. The same applies to results presented in Figure 2 where follow-up outcomes are presented without indication of the duration of follow-up. It is essential to present this information in a time-dependent manner (i.e., what percentage of patients discontinued over time; what proportion of patients were lost to follow-up after 6, 12 months; etc.…). Survival curves may be used.

We have now reported mortality and Loss to follow up as time dependent. Even if we plan a full paper analyzing the two outcomes.

  1. Overall viral suppression rate should be reported in the abstract as it is the primary outcome of this study.

This is done

  1. Lower likelihood of viral suppression was not observed in older adults but only in the 25-34 group. Thus, the conclusion page 7, in the abstract and in the first paragraph of the discussion needs to be corrected.

This is corrected

  1. The 1,000 copies cut-off used by the authors for high vs. low viral loads is not the most appropriate. They need to add more viral load categories in a sub-analysis to clarify their observation. As explained above, the following statement page 9: “… significantly lower in older patients…” is untrue. From data presented in Table 2, only the 25-34 group had significantly lower chances of non-suppression compared to 15-24 group. In addition, the youngest age group should be changed to 18-24 since the authors stated in the methods that inclusion was limited to adult above 18 years old. If minors were included, changes to the methods (including in ethical statements) should be made.

We made this edit

Minor comments:

  1. Page 2, in the description of the regimen, tenofovir should be replaced by tenofovir disoproxil fumarate.

We made this edit

  1. Page 2: “Some participants achieved…” This statement should be in the Results section rather than the Methods. Same for page 3 for the same statement.

This is done

  1. Page 2: “We used a semi-structured questionnaire…” The authors should specify that (I assume) the questionnaire was designed to gather sociological and ethnical data from the participants.

This is done

  1. Same section: “After pre-testing, the tools were…” This section is very vague. How many participants were enrolled in the pre-testing phase? What outcome was evaluated in this pre-testing phase (i.e., participation, satisfaction?). Do “tools” in fact mean questions included in the questionnaire?

We made these edits

  1. Page 3: “The primary objective…” The current study is an observational one. As such, the objective cannot be to achieve viral suppression (that would imply an interventional study). Rather the authors should indicate that their primary outcome was the proportion of patients with viral loads below 50 RNA copies per ml of plasma. Secondary outcome was identifying socio-economical and clinical factors associated with viral failure (as defined as viral load >50 RNA copies / ml at follow-up).

We made the edits in the text

  1. Can the authors comment on patients who self-reported as treatment naïve with viral suppression?

This was stated as follow: «Some patients self-declared naïve when they had an undetectable viral load. Some patients on ART who changed treatment sites did not disclose their treatment status (former ART case) to new providers for fear of being stigmatized. In a previous study, 19.4% of patients on ART did not disclose their status to family and friends [39]. It is difficult to testify about viral rebound among patients who self-declared naïve but had a viral load of less than 50 copies/mL and who ended up with a viral load of more than 50 copies after 6 months under the new regimen. »

  1. Figure 1 is useless as this information is already included in Table 1.

In Figure 1, we are making difference between naïve patients with with baseline VL < 50 copies/mL and naïve patients with baseline VL ≥ 50 copies/mL

  1. Page 5: “changes in viral load” are actually not presented in Table 2. It would be more accurate to state that baseline characteristics of individuals with available viral loads are presented in Table 2.

We made these edits

  1. Table 2. It may be better to present overall results at the top of the table.

We made these edits

  1. “Residence” should be moved to the left of the table.

We made these edits

  1. The definition for “normal” vs. “abnormal” serum creatinine levels should be clearly indicated in the methods.

We made these edits

  1. The authors need to discuss what in their opinion explains why individuals with higher creatinine levels were at higher chance of non-suppression. Is that because of discontinuation from medical indication? I expect not but the authors need to discuss this specific point rather than listing previous observations (that themselves do not provide an explanation for non-suppression).

We have added an explanation

  1. Page 9, second paragraph. The authors did not monitor/report side effects associated with dolutegravir use. Thus, the 2nd sentence should be edited as it refers to side effects.

We made these edits

  1. Page 9 “which removes viral load” is inaccurate. It would be better to state that dolutegravir-based therapy typically suppresses viral loads below limit of detection in most individuals within 6 months (and add references).

We made these edits

  1. Page 9, the sentence starting with “Whereas it is…” is incomplete. Please edit.

This was removed

Reviewer 2 Report

Buju and colleagues present an interesting study on the predictors of viral failure in HIV-positive individuals living in the context of an ongoing armed conflict in the city of Bunia, DRC. They enrolled 468 HIV-infected patients and measured the viral load 6-12 months after initiation of a DTG-based regimen. As a result, the authors observed that the likelihood of non-suppression was higher among individuals in advanced stages of disease, adolescents, individuals with abnormal serum creatinine, individuals with high baseline HIV viremia over 1000 copies/mL and individuals belonging to the Sudanese ethnic group. The authors conclude that more attention should be paid to ART-adherence in HIV-infected patients, especially among adolescents and economically vulnerable populations.

Overall, the study presents new interesting results, presenting real-life data on the success of a DTG-based regimen in a region of ongoing armed conflicts and shedding light on risk factors impeding the achievement of viral suppression. However, there are some comments which should be addressed before publication.

Specific comments

Introduction:

Page 1, line 3: It is “lamivudine” not “lamividine”.

Page 2, lines 7-8: The words “NNRTI” and PLHIV” were not introduced before using the abbreviations. The same for “CV” in line 18.

Page 2, line 9. Authors should write which regimen has been used until 2019.

Materials and Methods:

2.1: Page 3, lines 5-6: Why were pregnants and HBV-positive individuals excluded? Please give this information.

2.2: Page 3, lines 3-4 and page 4, lines 4-5: The information on patients achieving the viral suppression should be moved to the “results” part of the manuscript.

2.3: Page 3 line 1: instead of “presented” “analysed” should be used.

Results:

The authors should write the proportions consistently throughout the manuscript and should not change between x%, x.x% and x.xx%.

Page 3, line 7: The term “half of them” implicates that authors refer to the group before. However, as the total cohort is meant “Half of the participants” should be written.

Page 4, table 1: In total, both subgroups (naïve and experienced) are given with 100% although it must be n=177=37.8% and n=201=62.2%. Please correct this.

Page5, figures 1 and 2: Additionally, numbers should be given, not only the proportion.

Page 5, figure 1: Instead of “new” “naïve” should be used in the legend.

Page 5, figure 2: “Status of patients at the end of the study” would be a better description of the figure.

Page 6, table 2: Some numbers in the table are marked in bold. Are the numbers especially interesting? If so, it should be explained in the figure legend.

Page 6, table 2: The addendum “or indetectables” in the table caption is not necessary as in the following the authors do not discriminate between detectable and indetectable viral load in individuals with viral suppression.

Page 6, table 2: It is not clear what is meant exactly with “exposition time under DTG-based regimen in months”. Is this the time point when individuals achieved viral suppression? Or follow-up time? Please, explain more in detail.

Page 7, lines 4-5: Instead of writing “having VL of ≥50 copies/mL” two times in one sentence, authors should write “…with about 55% of these individuals failing viral suppression”.

Page 7, lines 4-5: Did the authors had a closer look on the 55% of Sudanese individuals with non-suppression? Is there any further correlation to high baseline viral load, stage of disease or ART-experience? Maybe this could shed more light on why the Sudanese groups has a higher risk of non-suppression if they e.g. already have a higher stage of disease due to being diagnosed later or other reasons.

Page 7, lines 7-9: Instead of “the worst” better write: “The viral response was significantly higher among participants at an advanced stage of the disease (Stage III and IV; 32.9%)”.

Page 7; lines 9-11: In table2, viral load results at 6–12 months are presented. However, the authors write that 31% “of self-reported naïve patients with a baseline VL of <50 copies/mL had a viral load over 50 copies/mL 6 months after initiation of the DTG-based regimen”. From the table it would be 31% of these individuals at 6-12 months after initiation. Please clarify.

Page 7, lines 13-15: Instead of “older patients” “adults” should be used throughout the manuscript

Discussion:

Page 8, line 3: “< 50 copies/mL” is meant according to table 2 (72,8% of individuals had viral suppression after 6-12 months).

Page 8, lines 5-6: Please give the non-suppression rate of the mentioned studies.

Page 8, 5-6: Is there any data on the non-suppression rate using non-DTG-based regimens in regions with armed conflicts for comparison?

Page 9, line 1: The term “SCr” should be explained before abbreviation.

Page 9, lines 23-24: Which figure?

Page 9, line 39-42: The authors do not know, at least from the information given in the text, if the individuals enrolled were in the acute phase.

Page 9, lines 44-45: Authors should write that the study of individuals not disclosing was performed in Nepal, which is probably a different setting from DRC.

Page 10, line 2: Only here, it becomes clear that “exposition time under DTG-based regimen in months” is equivalent with time-point of reaching viral suppression; see comment in the results section. It would be best to explain this already in the Materials&Methods section.

Page 10, line 2: What have “resource constraints” to do with time-point of individuals reaching VL-suppression?

Another limitation of the study is the low number of individuals from the Sudanese ethnic group. Authors therefore should write, that these results should be evaluated with caution and more individuals of this ethnic group should be analysed to support their results.

Page 10, line 5: “LTFU” is not introduced before using the abbreviation. 

Author Response

REVIEWER 2:

Specific comments

Introduction:

  1. Page 1, line 3: It is “lamivudine” not “lamividine”.

This is done

  1. Page 2, lines 7-8: The words “NNRTI” and PLHIV” were not introduced before using the abbreviations. The same for “CV” in line 18.

We made these edits

  1. Page 2, line 9. Authors should write which regimen has been used until 2019.

This is done

Materials and Methods:

  1. 1: Page 3, lines 5-6: Why were pregnants and HBV-positive individuals excluded? Please give this information.

This was removed. This was not for this study but another one we are conducting, This was a mistake

  1. 2: Page 3, lines 3-4 and page 4, lines 4-5: The information on patients achieving the viral suppression should be moved to the “results” part of the manuscript.

This is done

  1. 3: Page 3 line 1: instead of “presented” “analysed” should be used.

This is done

Results:

  1. The authors should write the proportions consistently throughout the manuscript and should not change between x%, x.x% and x.xx%.

We made these edits

  1. Page 3, line 7: The term “half of them” implicates that authors refer to the group before. However, as the total cohort is meant “Half of the participants” should be written.

We made these edits

  1. Page 4, table 1: In total, both subgroups (naïve and experienced) are given with 100% although it must be n=177=37.8% and n=201=62.2%. Please correct this.

We made these edits

  1. Page5, figures 1 and 2: Additionally, numbers should be given, not only the proportion.

We made these edits

  1. Page 5, figure 1: Instead of “new” “naïve” should be used in the legend.

We made these edits

  1. Page 5, figure 2: “Status of patients at the end of the study” would be a better description of the figure.

We made these edits

  1. Page 6, table 2: Some numbers in the table are marked in bold. Are the numbers especially interesting? If so, it should be explained in the figure legend.

We have edited accordingly

  1. Page 6, table 2: The addendum “or indetectables” in the table caption is not necessary as in the following the authors do not discriminate between detectable and indetectable viral load in individuals with viral suppression.

We have edited accordingly

  1. Page 6, table 2: It is not clear what is meant exactly with “exposition time under DTG-based regimen in months”. Is this the time point when individuals achieved viral suppression? Or follow-up time? Please, explain more in detail.

Exposition time under DTG-based regimen in months, is the duration from initiation of the DTG-based regimen to the day when collecting blood sample to assess viral load.

  1. Page 7, lines 4-5: Instead of writing “having VL of ≥50 copies/mL” two times in one sentence, authors should write “…with about 55% of these individuals failing viral suppression”.

We have edited accordingly

  1. Page 7, lines 4-5: Did the authors had a closer look on the 55% of Sudanese individuals with non-suppression? Is there any further correlation to high baseline viral load, stage of disease or ART-experience? Maybe this could shed more light on why the Sudanese groups has a higher risk of non-suppression if they e.g. already have a higher stage of disease due to being diagnosed later or other reasons.

We did compare Sudanese and other ethnics groups, even if we have small number of Sudanese group, we did found similar distribution according WHO Stage of disease, viral load at baseline and Status of treatment at baseline. Seethe below table:

N

Others ethnic groups

Sudanese group

p-value

n

%

n

%

Stage

0.652

I and II

219

203

47.1

16

43.2

III and IV

249

228

52.9

21

56.8

Viral load at baseline

0.318

<1000 copies/mL

313

291

67.5

22

59.5

≥1000 copies/mL

155

140

32.5

15

40.5

Status of treatment at baseline

0.722

Self-reported naïve patients

177

162

37.6

15

40.5

Under ART (Experienced)

291

269

62.4

22

59.5

Total*

468

431

92.1

37

7.9

  1. Page 7, lines 7-9: Instead of “the worst” better write: “The viral response was significantly higher among participants at an advanced stage of the disease (Stage III and IV; 32.9%)”.

We made these edits accordingly, and this can be seen as follow in the current version: «The viral response was significantly lower among participants at an advanced stage of the disease (Stage III and IV; 32.9% of non-suppression) ».

  1. Page 7; lines 9-11: In table2, viral load results at 6–12 months are presented. However, the authors write that 31% “of self-reported naïve patients with a baseline VL of <50 copies/mL had a viral load over 50 copies/mL 6 months after initiation of the DTG-based regimen”. From the table it would be 31% of these individuals at 6-12 months after initiation. Please clarify.

We made these edits accordingly

  1. Page 7, lines 13-15: Instead of “older patients” “adults” should be used throughout the manuscript

We made these edits accordingly

Discussion:

  1. Page 8, line 3: “< 50 copies/mL” is meant according to table 2 (72,8% of individuals had viral suppression after 6-12 months).

We made these edits accordingly

  1. Page 8, lines 5-6: Please give the non-suppression rate of the mentioned studies.

We made these edits accordingly

  1. Page 8, 5-6: Is there any data on the non-suppression rate using non-DTG-based regimens in regions with armed conflicts for comparison?

No, We did not find them.

  1. Page 9, line 1: The term “SCr” should be explained before abbreviation.

This is done

  1. Page 9, lines 23-24: Which figure?

We made this edits: Results instead of figure

  1. Page 9, line 39-42: The authors do not know, at least from the information given in the text, if the individuals enrolled were in the acute phase.

No, we do not have this information.

  1. Page 9, lines 44-45: Authors should write that the study of individuals not disclosing was performed in Nepal, which is probably a different setting from DRC.

Thanks so much, we stated it in the current version.

  1. Page 10, line 2: Only here, it becomes clear that “exposition time under DTG-based regimen in months” is equivalent with time-point of reaching viral suppression; see comment in the results section. It would be best to explain this already in the Materials&Methods section.

This is done

  1. Page 10, line 2: What have “resource constraints” to do with time-point of individuals reaching VL-suppression?

We would have liked the collection of the blood sample to assess viral load after initiation of the DTG-based regimen to be performed after a fixed duration of exposure (six months) for all participants. 

  1. Another limitation of the study is the low number of individuals from the Sudanese ethnic group. Authors therefore should write, that these results should be evaluated with caution and more individuals of this ethnic group should be analysed to support their results.

Thanks so much, we stated it in the current version.

  1. Page 10, line 5: “LTFU” is not introduced before using the abbreviation. 

This is done